

# Fine grained compositional analysis of Port Everglades Inlet microbiome using high throughput DNA sequencing

Lauren O'Connell[1,*], Song Gao[2,*], Donald McCorquodale[3], Jay Fleisher[4] and Jose V. Lopez[1,*]

[1] Department of Biological Sciences, Halmos College of Natural Sciences and Oceanography, Nova Southeastern University, Dania Beach, FL, United States of America

[2] Division of Natural Sciences, Duke Kunshan University, Kunshan, Jiangsu, China

[3] Department of Marine and Environmental Sciences, Halmos College of Natural Sciences and Oceanography, Nova Southeastern University, Dania Beach, FL, United States of America

[4] College of Osteopathic Medicine, Nova Southeastern University, Davie, FL, United States of America

[*] These authors contributed equally to this work.

Corresponding author
Lauren O'Connell,
locon833@gmail.com

## ABSTRACT

**Background**. Similar to natural rivers, manmade inlets connect inland runoff to the ocean. Port Everglades Inlet (PEI) is a busy cargo and cruise ship port in South Florida, which can act as a source of pollution to surrounding beaches and offshore coral reefs. Understanding the composition and fluctuations of bacterioplankton communities ("microbiomes") in major port inlets is important due to potential impacts on surrounding environments. We hypothesize seasonal microbial fluctuations, which were profiled by high throughput 16S rRNA amplicon sequencing and analysis.

**Methods & Results**. Surface water samples were collected every week for one year. A total of four samples per month, two from each sampling location, were used for statistical analysis creating a high sampling frequency and finer sampling scale than previous inlet microbiome studies. We observed significant differences in community alpha diversity between months and seasons. Analysis of composition of microbiomes (ANCOM) tests were run in QIIME 2 at genus level taxonomic classification to determine which genera were differentially abundant between seasons and months. Beta diversity results yielded significant differences in PEI community composition in regard to month, season, water temperature, and salinity. Analysis of potentially pathogenic genera showed presence of *Staphylococcus* and *Streptococcus*. However, statistical analysis indicated that these organisms were not present in significantly high abundances throughout the year or between seasons.

**Discussion**. Significant differences in alpha diversity were observed when comparing microbial communities with respect to time. This observation stems from the high community evenness and low community richness in August. This indicates that only a few organisms dominated the community during this month. August had lower than average rainfall levels for a wet season, which may have contributed to less runoff, and fewer bacterial groups introduced into the port surface waters. Bacterioplankton beta diversity differed significantly by month, season, water temperature, and salinity. The 2013–2014 dry season (October–April), was warmer and wetter than historical averages. This may have driven significant differences in beta diversity. Increased nitrogen and phosphorous concentrations were observed in these dry season months,

possibly creating favorable bacterial growth conditions. Potentially pathogenic genera were present in the PEI. However their relatively low, non-significant abundance levels highlight their relatively low risk for public health concerns. This study represents the first to sample a large port at this sampling scale and sequencing depth. These data can help establish the inlet microbial community baseline and supplement the vital monitoring of local marine and recreational environments, all the more poignant in context of local reef disease outbreaks and worldwide coral reef collapse in wake of a harsh 2014–16 El Niño event.

## INTRODUCTION

A continental coastal zone can represent a gradient across distinct biogeographical boundaries (freshwater, brackish and saltwater). Mangroves, streams, or manmade inlets in these transition zones provide potential links. Fort Lauderdale's Port Everglades Inlet (PEI) (also known as the Port Everglades Shipping Channel [PESC]), in Broward County, FL is a man-made, dredged, deep-water port located along the southeastern coast of subtropical Florida (*Stauble, 1993*; http://www.porteverglades.net; *NOS, 2011*). Located directly offshore from the PEI is a major US coral reef tract (*Staley et al., 2017*), as well as multiple recreational beaches, fishing piers, and watersport areas (*Stamates et al., 2013*). Coral reefs, beaches, and recreational water sport areas are impacted both positively and negatively by resident microbial communities of these areas. The ecosystem services of bacteria in marine communities include nutrient cycling and symbiosis, while disadvantages include the possible presence of marine pathogens, which may cause illness in the marine environment or to humans utilizing it.

Although large-scale bacterioplankton community (or "microbiome") studies such as the Tara expeditions or the Earth Microbiome project, have their place providing important global information on a diverse array of marine microbial habitats (*Sunagawa et al., 2015*; *Thompson et al., 2017*), much value can also be obtained by focusing on finer scale microbial dynamics of distinct local habitats especially when unique. PEI is one of largest cargo and cruise ship ports in the country and is susceptible to pollution and pathogens from cargo and cruise ships via ballast water flushing, oil leakage, and cargo spills, and through land-based pollution. The past decade has marked rapid increases in human population and coastal development, resulting in the input of human pathogens and enteric bacteria into coastal waters (*Mallin et al., 2000*; *Lapointe et al., 2015*). The introduction of these pathogens can cause human illnesses such as skin, ear, and gastric infections and can cause disease in marine organisms. This was seen with the introduction of the human pathogen *Serratia marcescens,* into a coral reef ecosystem, which was found to be the causative agent in the bleaching event seen in the Carribbean elkhorn coral (*Sutherland et al., 2011*). Pollution into PEI also occurs via land-based runoff. Land-based runoff contributes to nutrient, pathogen, and pollutant deposition into the marine
environment, which can lead to eutrophication (*Futch et al., 2011*; *Lapointe et al., 2015*; *Staley et al., 2017*). Eutrophication leads to conditions of hypoxia and anoxia and decreased biodiversity causing major shifts in community structure of an ecosystem. These shifts can result in an increase of phytoplankton, which can cause harmful algal blooms, and can eventually lead to reduction of zooplankton and bacterioplankton diversity (*Marcus, 2004*; *Paerl et al., 2003*). Southern Florida has two main seasons. The wet season mostly ranges from May–September, while the dry season occurs from October–April. The wet season is mostly during the hot summer months, and is characterized by increased rainfall levels and increased temperatures. With increased rainfall levels, also comes increased runoff into the marine environment. Previous research has indicated that influxes of nutrients from agriculture practices in near-shore environments are contributing to degradation of Florida reef ecosystems via eutrophication and introduction of pathogens (*Finkl & Charlier, 2003*; *Banks et al., 2008*; *Walker, Riegl & Dodge, 2008*; *Lewis et al., 2017*).

This study reports an extensive environmental next-generation sequencing characterization of the bacterioplankton microbiome from the surface seawater in PEI. Water samples were collected from June 2013 to May 2014 to determine monthly and seasonal alpha ($\alpha$) and beta ($\beta$) diversity fluctuations. Fluctuations of the bacterioplankton community in PEI's surface water was monitored over a year. While samples were collected weekly, a total of four samples per month, two from each location in the port, were used for consistency in statistical analyses. This study provides a higher sampling resolution than has been currently reported. To date, inlet study data has been collected bi-monthly, quarterly, and for a shorter sampling periods (*Wallace et al., 2018*; *Staley et al., 2017*; *Campbell et al., 2015*).

The primary hypothesis of this study predicted strong seasonal effects on the inlet microbiome in PEI due to temperature and rainfall. It was also hypothesized that potentially harmful pathogens to both humans would be present in a higher abundance during the wet season.

This study applied high-throughput (HTP) sequencing technology to complete DNA sequencing of surface seawater samples in PEI, extending from previous inlet studies, which were largely restricted to a smaller sample sizes, lower sampling frequency, and culture-based or RT-qPCR methods (*Symonds et al., 2016*; *Aranda et al., 2015*; *Campbell et al., 2015*).

## MATERIALS & METHODS

### Water sample collection, DNA extraction, and chemical analysis

A total of 151 surface seawater samples were collected weekly, at ebb tide, from PEI in Broward County, FL by kayak over a year-long timespan (June 2013–July 2014). Three 1.0 liter water samples were collected at a depth of 0.5 m every week from two different sites within the inlet. Water temperature was measured *in situ* at time of sampling with a glass thermometer. Salinity measurements were taking immediately upon returning to lab (within 30 min of sample collection) with a refractometer. Precipitation values were obtained using NOAAs data from the National Center for Environmental Information (http://www.ncdc.noaa.gov/cag/time-series). For each site, 1.0 L of water was filtered using Pall GN Metricel® grid 47 mm, 0.45μm filters, by vacuum filtration.

Total microbial genomic DNA was extracted using MO BIO's PowerLyzer™ PowerSoil® kit (Carlsbad, CA, USA). Genomic DNA size and quality was verified with a 1% agarose gel and Nanodrop measurements. A total of 81 samples were queued for sequencing based on quality of extracted DNA. Samples that showed clear, non-degraded, high molecular weight genomic DNA bands and had 260/280 ratios ~1.8 were considered quality DNA. After gel verification, DNA concentration was measured using the Qubit 2.0 (Life Technologies, Carlsbad, CA, USA).

Surface water samples collected at each site were subjected to ion chromatography (IC) analysis using a Thermo Scientific Dionex ICS-1600 system (Bannockburn, IL, USA). After filtration of particulates using syringe filters, samples were diluted 1,000 times before injection into the IC. Ion chromatography analysis was used to detect the presence and measure the concentrations of chemical ions in the PEI surface water. A total of five anions—fluoride, chloride, nitrate, phosphate, and sulfate—were analyzed with calibration curves from standard solutions and detection limits at approximately 10 ppb.

## Sequencing preparation

Samples were prepared for MiSeq® sequencing following Illumina's 16S Metagenomic Sequencing Library Preparation guide (*Illumina, 2013*). The amplicon primers used for the first PCR were universal primers, MIDf-515F and 806rc which amplified the V4 region of the 16S rRNA gene (*Caporaso et al., 2010a*; *Caporaso et al., 2010b*). Illumina adaptor sequences were added to these primers as outlined in the aforementioned Metagenomic Sequencing Library Preparation Guide (*Illumina, 2013*). A second PCR was completed to attach Illumina Nextera indices using the Illumina Nextera indexing kit so that samples could be demultiplexed (*Illumina, 2013*). The final pooled DNA library was diluted to a concentration of 4 pM with a 50% spike in of 12.5 pM PhiX.

## Sequence analyses

Sequence analysis was performed using Quantitative Insights into Microbial Ecology 2 (QIIME2) (*Caporaso et al., 2010a*). Paired ends were joined using PANDAseq 2.8.1 at a 90% confidence level (*Masella et al., 2012*). A total of 4 samples per month, 2 from each location, resulting in a total of 48 samples were used for downstream analysis so as not to skew statistical analysis. Paired end sequences were imported into QIIME2 and sequence quality control was completed using the DADA2 pipeline incorporated into QIIME. The DADA2 program filtered out phiX reads, removed chimeric sequences, and assigned reads into Operational Taxonomic Units (OTUs). The sequences were truncated to 220 basepairs because the quality of reverse reads after base 220 declined. OTUs were grouped into clusters at 97% sequence similarity (*Callahan et al., 2016*). Taxonomy was assigned using the DADA2 pipeline which implemented the Ribosomal Database Project (RDP) naïve Bayesian classifier (*Callahan et al., 2016*).

## Statistical analyses

All statistical analyses were completed using QIIME2 and R Statistical Software Version 1.1.383. The R package phyloseq was used to generate stacked bar charts depicting relative abundance of taxa (*McMurdie & Holmes, 2013*). Richness and evenness estimates were

calculated using QIIME2 (https://qiime2.org). To determine statistical significance in alpha diversity non-parametric Kruskal–Wallis tests using Pielou's evenness values were used to complete pairwise comparisons between month, location, and weather at time of sampling (rain vs. no rain). Non-parametric Kruskal–Wallis comparisons were performed using Faith's Phylogenetic Diversity to determine if there were statistical differences in species richness in the PEI samples. Box and whisker plots for species richness and evenness were generated using QIIME2. To test non-discrete metadata (salinity and water temperature) for correlations with alpha diversity, Spearman's Rank correlation analysis was completed using Pielou's evenness values. Taxonomic stacked bar charts were generated using data from QIIME2 imported into Phyloseq. Analysis of composition of microbiomes (ANCOM) test were run to determine if there were significant differences in the relative abundance of any taxa between month or season (*Mandal et al., 2015*).

Statistical analyses for beta diversity were completed by calculating Bray–Curtis distance using QIIME2. A PERMANOVA was run to complete pairwise comparisons of samples for month, location, and season. Principle Coordinate Analysis (PCoA) plots were generated for month sample taken and for season from Bray–Curtis distances using QIIME2. A Mantel test was run to determine if there were statistically significant correlations in beta diversity in relation to non-discrete metadata.

### Regression analysis

A series of Multiple Least Squares Regressions were used to assess possible relationships between each bacterial class and the environmental variables taken as part of the study. The $R^2$ value is the measure used to determine how well the data fits the regression line. The higher $R^2$ values indicated better data fits with the model. The class level taxonomic classification was used to evaluate relationships between environmental variables because majority of the sequences were classified to the class level. A backward selection method was used with both entry and model retention. All regression analyses were carried out using SAS Statistical Software (SAS Institute, Cary, NC, USA).

### Potential pathogen detection

Potentially pathogenic bacteria were detected by filtering out genera that may be pathogenic. Statistical analysis was completed on *Staphylococcus* and *Streptococcus spp.* using the ANCOM test. This test compares abundances of genera to determine if they change significantly between populations or environments (*Mandal et al., 2015*). To display presence of potential pathogens graphically, the relative abundance, genus, and month were plotted and faceted by season using the plot_bar command in phyloseq.

## RESULTS

### 16S rRNA sequence output overview

A total of 151 samples were collected weekly from PEI from July 2013–June 2014; among them 81 samples were queued for DNA sequencing. A total of 48 samples, four per month, two from each site, were used for downstream bioinformatics analysis (Table S1). The MiSeq sequencing yielded a total of 1,435,072 raw sequences with Q scores greater than

30 and an average read length of 255 basepairs. The average number of reads per sample was 20,586 with the minimum number of reads per sample being 9,199 and the maximum number of reads per sample being 35,108. A total of 25,020 chimeric sequences were removed from the dataset with a total of 1,410,052 sequences left for OTU table generation and database alignment. After filtering of sequences to remove identical sequences and subsequences a total of 395,009 unique sequences were left for taxonomic assignment using the Greengenes 13_8 database. A total of 16,384 OTUs were generated.

## Alpha diversity

Alpha diversity was assessed through species evenness and species richness measures. Results for species evenness were statistically significant for month and season ($p = 0.015$; $p = 0.020$ respectively). There were no significant differences in species evenness for PEI samples by location, or weather at time of sampling (raining vs. not raining) ($p > 0.05$). The month of August had the lowest species evenness values indicating that these samples were dominated only by a few organisms. The month of December had the highest level of species evenness (Fig. S1). The wet season has lower species evenness values than the dry season (Fig. 1). In contrast, no significant differences in alpha diversity appeared in relation to water temperature or salinity ($p > 0.05$).

Statistically significant differences were seen in species richness in relation to month and season ($p = 0.010$; $p = 0.025$ respectively). For example, the month of December had the lowest measure of species richness, while September had the highest measure of species richness (Fig. S2). Concordantly, the wet season samples had higher species richness values than the dry season (Fig. 1). In contrast, species richness analyses across samples indicated that there were no statistically significant differences in species richness in relation to rain at time of sampling versus no rain at time of sampling, water temperature values, salinity measures, or sampling location ($p > 0.05$).

## Beta diversity

The PEI samples were analyzed to determine if there were significant differences in the microbial communities due to differences in sampling location, month that the sample was taken, season, weather at time of sampling, salinity, and water temperature.

**Location**. The results of the PERMANOVA test using Bray–Curtis dissimilarity indicated no significant differences in beta diversity of samples taken at two different locations in PEI ($p = 0.99$).

**Weather at Time of Sampling (Raining vs. Not Raining)**. There were significant differences in beta diversity of PEI samples in regards to weather (rain vs. no rain) at the time of sampling ($p = 0.03$).

**Season**. Results of the PERMANOVA indicated that there were significant differences in the microbial community composition between seasons ($p = 0.001$) There is separation of points in the PCoA plot between the wet and dry seasons, displaying that there are differences in the microbial community composition (Fig. 2).

**Month**. The results of the PERMANOVA test using Bray–Curtis Dissimilarity indicated that there were overall significant differences observed in the beta diversity of all months tested ($p = 0.001$). Multiple pairwise comparisons were completed to determine which
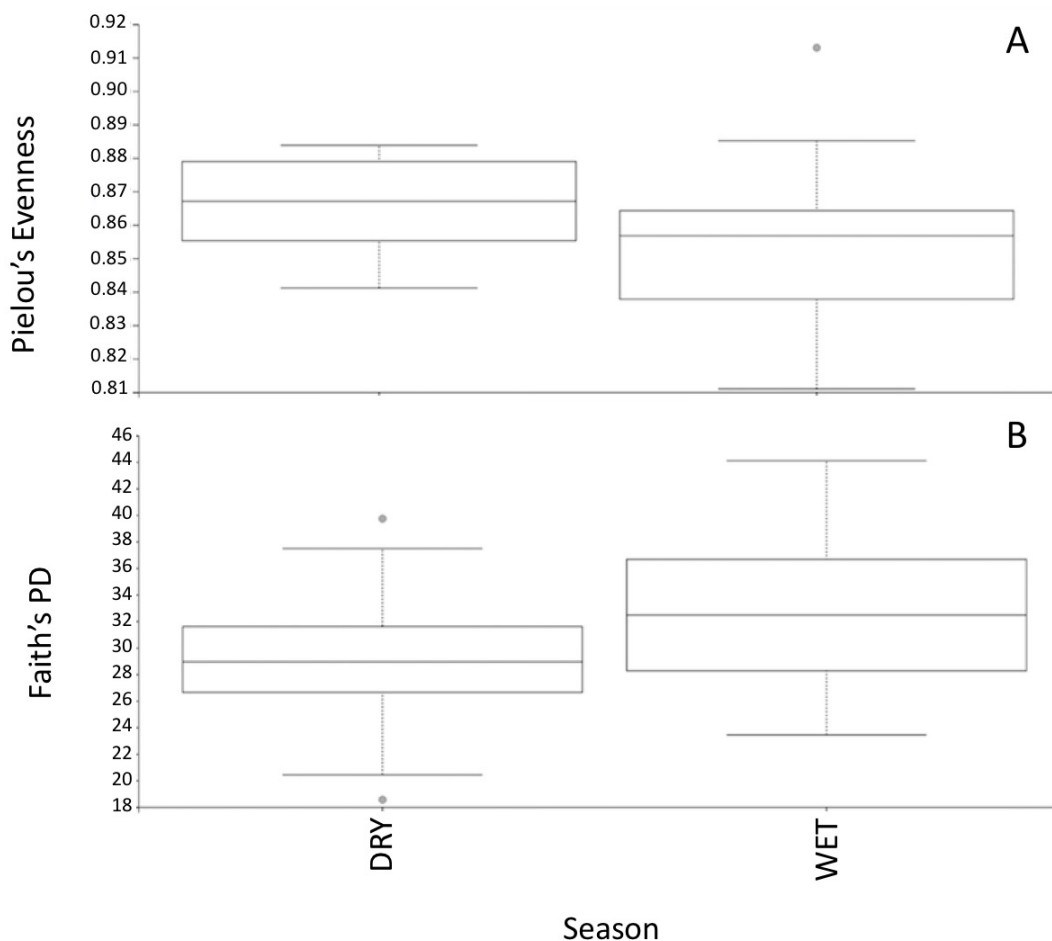

**Figure 1 Alpha diversity measurements of PEI microbiome.** (A) Comparison of species evenness in PEI between seasons. Pielou's Evenness values were used to calculate species evenness. Significant differences were observed between seasons of PEI for species evenness, with the dry season having lower species evenness ($p = 0.020$). (B) The comparison of species richness in PEI between seasons. Faith's Phylogenetic Diversity values were used to calculate species richness. Significant differences were observed between seasons of PEI for species richness, with the wet season having higher species richness values ($p = 0.025$).

months differed significantly from one another. All months differed significantly in beta diversity except for August and July, February and March, February and January, July and September, July and October, and July and June (Table S2).

### Abundant genera

To better understand how the microbial community composition fluctuated with season we examined which organisms were present at different taxonomic levels and their relative abundance levels. Proteobacteria, Planctomycetes, Cyanobacteria, Bacteroidetes, Actinobacteria, Verrucomicrobia, Tenericutes, and Euryarchaeota were the most abundant phyla across all samples, with Proteobacteria making up at least 25% of the bacterial community in all samples (Fig. S3). At the lower taxonomic levels, an increased number of taxa were observed in wet season in comparison to the dry season (13 vs 11 taxa) (Fig. 3).
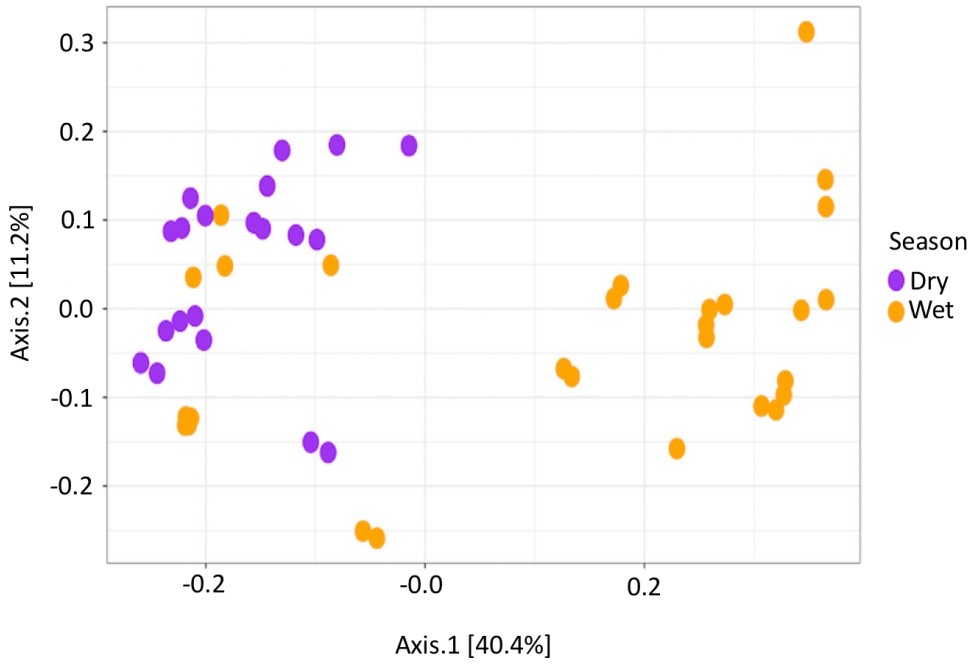

**Figure 2 Beta diversity measurements of PEI microbiome.** Principle Coordinate Analysis of PEI water samples using Bray–Curtis distance. Clustering of points indicates sample similarity. Significant differences in beta diversity were observed between the wet and the dry season ($p = 0.001$).

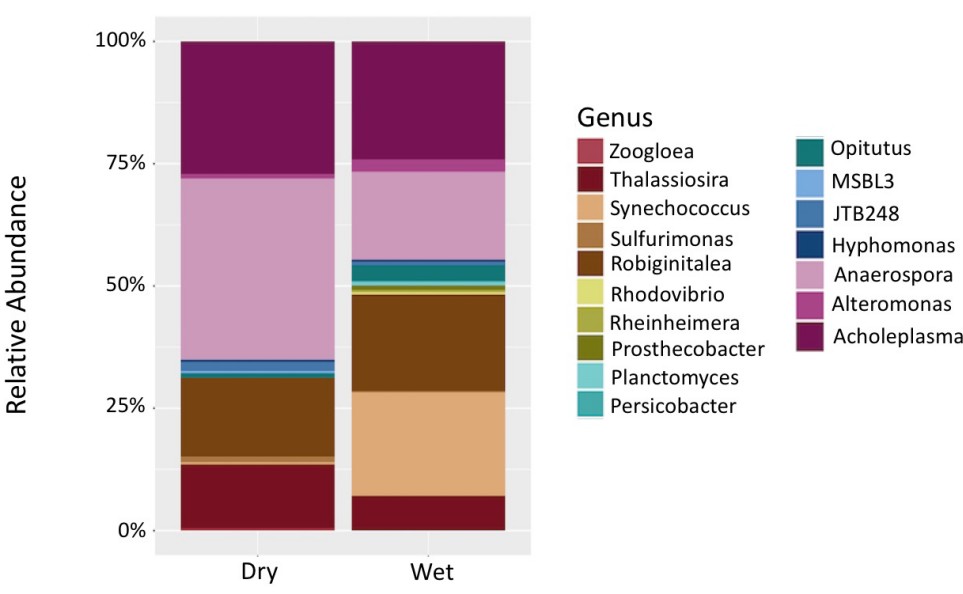

**Figure 3 Comparison of relative abundance values of genus-level taxa between seasons.** Stacked bar chart displaying seasonal fluctuations of relative abundance levels at genus-level classification. *Synechococcus* was the only taxon to have significantly different abundance levels between the wet and the dry season.

While observable fluctuations in relative abundance levels are seen in the *Opitutus* and *Anaerospora* genera, results from the ANCOM tests indicated that at the genus level, only *Synechococcus* differed significantly in its abundance levels from the dry to the wet season. *Synechococcus spp.* went through large increases in abundance levels from the dry season to the wet season (Table 1).

## Linear regression analysis with chemical and environmental metadata

Multiple least Squares regression analysis results showed that there were statistically significant relationships between certain taxa and salinity measures, chloride ion concentration, water temperature, sulfate ion concentration, and rainfall (Table 2). An interesting trend to point out in the data was that *Proteobacteria* and *Cyanobacteria* showed an inverse trend in abundance, which correlated with temperature and rainfall data to some degree (Fig. S4). Environmental metadata for average precipitation levels and temperature was obtained through NOAA National Centers for Environmental information (*NOAA, 2018*). Rainfall averages for the month of August were 3.4 inches lower than average historical data. The remaining wet season months (May–October) generally had higher than average rainfall levels (Fig. 4). The average temperature during the wet season months was generally higher than the historical average, while the winter months had lower than average temperatures in comparison to historical data (Fig. 4).

## Potentially Pathogenic Bacteria

Results of the ANCOM tests demonstrated that there were no significant differences in the relative abundance levels of potential pathogens during the year (Tables S3; S4). In the *Bacillales, Clostridiales,* and *Lactobacillales* orders, the most abundant potentially pathogenic genera were *Staphylococcus* and *Streptococcus. Staphylococcus* was in highest relative abundance during the summer months of July and August, though this increase in abundance was not statistically significant (Fig. 5). *Staphylococcus* was also present during April and March, but this increase in abundance was not statistically significant. *Streptococcus* was present in much lower abundance than *Staphylococcus sp.* It was present in its highest abundance in April and was too low to be detected in other months. This increase in abundance was not statistically significant (Fig. 5).

## DISCUSSION

Inlets are not rivers, but both share several similarities: large volumes of water flowing out of them; connection of inland city and agriculture runoff with the ocean; and generation of visible plumes distinct from offshore marine waters. However, as a mostly artificial construction, inlets may be subject to more detailed or controlled characterizations. Our study considered that different water masses (shallow and deeper) may be generated during ebb tides, which can lead to hydrodynamic complexity (*Stamates et al., 2013*). For example, the mean volume of ebb tide surface water flow (4.41 million cubic meters per tidal phase) through the PEI is about half the water flow of the deep channel. Although our sampling site was slightly outside the Intracoastal waterway (ICW), we expected our water samples

O'Connell et al. (2018), *PeerJ*, DOI 10.7717/peerj.4671

**Table 1   ANCOM test results for differential abundance of *Synechococcus spp*.** The ANCOM results for *Synechococcus spp*. that differed significantly in abundance levels between seasons. High *W* values indicate significant differences in abundance levels between seasons. The higher the *W* value the more significant the differences in abundance levels between seasons.

| Percentile | 0 | 25 | 50 | 75 | 100 | 0 | 25 | 50 | 75 | 100 | | |
|---|---|---|---|---|---|---|---|---|---|---|---|---|
| OTU | Dry | Dry | Dry | Dry | Dry | Wet | Wet | Wet | Wet | Wet | Reject null hypothesis | *W* |
| Unidentified *Synechococcus sp*. 1 | 1 | 1 | 42.5 | 60.5 | 159 | 31 | 86.25 | 120 | 159.75 | 313 | TRUE** | 14 |
| Unidentified *Synechococcus sp*. 3 | 1 | 1 | 1 | 1 | 11 | 1 | 1 | 1 | 1 | 21 | TRUE** | 13 |
| Unidentified *Synechococcus sp*. 5 | 1 | 1 | 1 | 8 | 57 | 1 | 23.75 | 41.5 | 79 | 132 | TRUE** | 14 |
| Unidentified *Synechococcus sp*. 7 | 86 | 160.25 | 378 | 539.5 | 2,172 | 231 | 822 | 1,725 | 2,411.25 | 4,189 | TRUE** | 11 |
| Unidentified *Synechococcus sp*. 8 | 1 | 1 | 1 | 1 | 34 | 1 | 1 | 1 | 1 | 57 | TRUE** | 13 |
| Unidentified *Synechococcus sp*. 9 | 1 | 1 | 1 | 1 | 1 | 1 | 1 | 1 | 1 | 25 | TRUE** | 13 |
| Unidentified *Synechococcus sp*. 11 | 1 | 1 | 1 | 1 | 1 | 1 | 1 | 1 | 1 | 27 | TRUE** | 13 |
| Unidentified *Synechococcus sp*. 13 | 1 | 1 | 1 | 1 | 1 | 1 | 1 | 26.5 | 41.75 | 114 | TRUE** | 14 |
| Unidentified *Synechococcus sp*. 15 | 1 | 1 | 1 | 1 | 1 | 1 | 1 | 51.5 | 123.25 | 237 | TRUE** | 18 |
| Unidentified *Synechococcus sp*. 18 | 1 | 1 | 1 | 1 | 1 | 1 | 1 | 12.5 | 51 | 98 | TRUE** | 11 |
| Unidentified *Synechococcus sp*. 20 | 1 | 1 | 1 | 1 | 1 | 1 | 1 | 32.5 | 64.25 | 213 | TRUE** | 15 |
| Unidentified *Synechococcus sp*. 21 | 97 | 164.25 | 202.5 | 292.75 | 539 | 44 | 129.25 | 179 | 301.75 | 752 | TRUE** | 13 |
| Unidentified *Synechococcus sp*. 24 | 1 | 1 | 1 | 1 | 1 | 1 | 1 | 30.5 | 47.75 | 105 | TRUE** | 16 |
| Unidentified *Synechococcus sp*. 25 | 1 | 1 | 1 | 1 | 29 | 1 | 1 | 1 | 1 | 1 | TRUE** | 14 |
| Unidentified *Synechococcus sp*. 29 | 1 | 1 | 1 | 1 | 1 | 1 | 1 | 1 | 1 | 19 | TRUE** | 13 |

**Notes.**
** Indicates statistical significance.

**Table 2** **Linear regression analysis of environmental metadata in relation to microbial taxa observed.** Multiple least squares regression analysis at class level taxonomy. Gammproteobacteria, Alphaproteobacteria, & Betaproteobacteria were abbreviated for space. There were statistically significant relationships with specific taxa and salinity, water temperature, rainfall, and chloride ion concentration. Alpha = 0.10.

| | Gamma | Flavobacteriia | Acidomicrobiia | Alpha | Beta | Synechococcophy cideae | Actinobacteria | Unclassified |
|---|---|---|---|---|---|---|---|---|
| R-Squared | 0.21 | 0.21 | 0.21 | 0.13 | 0.14 | 0.2 | 0.21 | 0.07 |
| Salinity | 0.017[**] | 0.101 | 0.0013[**] | 0.0008[**] | 0.0005[**] | NA | 0.051[**] | NA |
| Water Temperature | 0.060[**] | 0.020[**] | NA | NA | NA | 0.001[**] | 0.0001[**] | NA |
| Chloride | NA | NA | NA | NA | NA | 0.016[**] | NA | NA |
| Sulfate | NA | NA | NA | NA | NA | NA | NA | NA |
| Rainfall | NA | NA | NA | NA | NA | NA | NA | 0.0184[**] |

**Notes.**
[**]Indicates statistical significance.

to more closely resemble a brackish surface water microbial community rather than an open ocean water microbial community. This was expected as PEI is influenced by human development and is subject to land-based pollution, which is not as common in open ocean waters. Beach samples and oceanic samples would need to be collected and compared to see which of these environments the inlet most resembles. In a study by Miller and colleagues (*2016*), it was found that dredging of the Port of Miami from late 2013 to early 2015 increased coral tissue loss on adjacent reefs due to sedimentation. The dredging in the Port of Miami appears to have exacerbated local environmental stress due to the coincidence with increased water temperatures leading to a mass bleaching event and increases in reef disease (*Miller et al., 2016*). This points to the utility of the present report as a baseline prior to the PEI dredging slated to occur in the near future.

An earlier study from our laboratory (*Campbell et al., 2015*) utilized high-throughput 454-pyrosequencing technology to generate baseline knowledge of PEI microbiomes, but was limited to sampling quarterly. In the current study multiple samples per month were analyzed to assess variation and generate a more complete baseline of the microbial community present in PEI. Utilization of Illumina MiSeq sequencing technology also provided more reads per sample, and therefore more coverage, adding confidence to taxonomic assignments. Both studies on the PEI microbiome yielded complementary data creating a strong baseline of the microbiome present in the inlet. These studies can be used by local county and public health officials, who conduct routine monitoring on port waters, and by environmental scientists looking to see what the impacts of the microbial community in PEI might have on the surrounding coastal beaches and the adjacent Florida coral reef (*Aranda et al., 2015*).

## Bacterioplankton community composition fluctuations: location, month, and season

Seasonal and monthly diversity fluctuations were observed for the PEI samples. The dry season had much higher species evenness, but lower species richness when compared to the wet season. This means that the communities in the dry season were dominated by a few organisms in relatively equal abundances, while during the wet season more taxa were

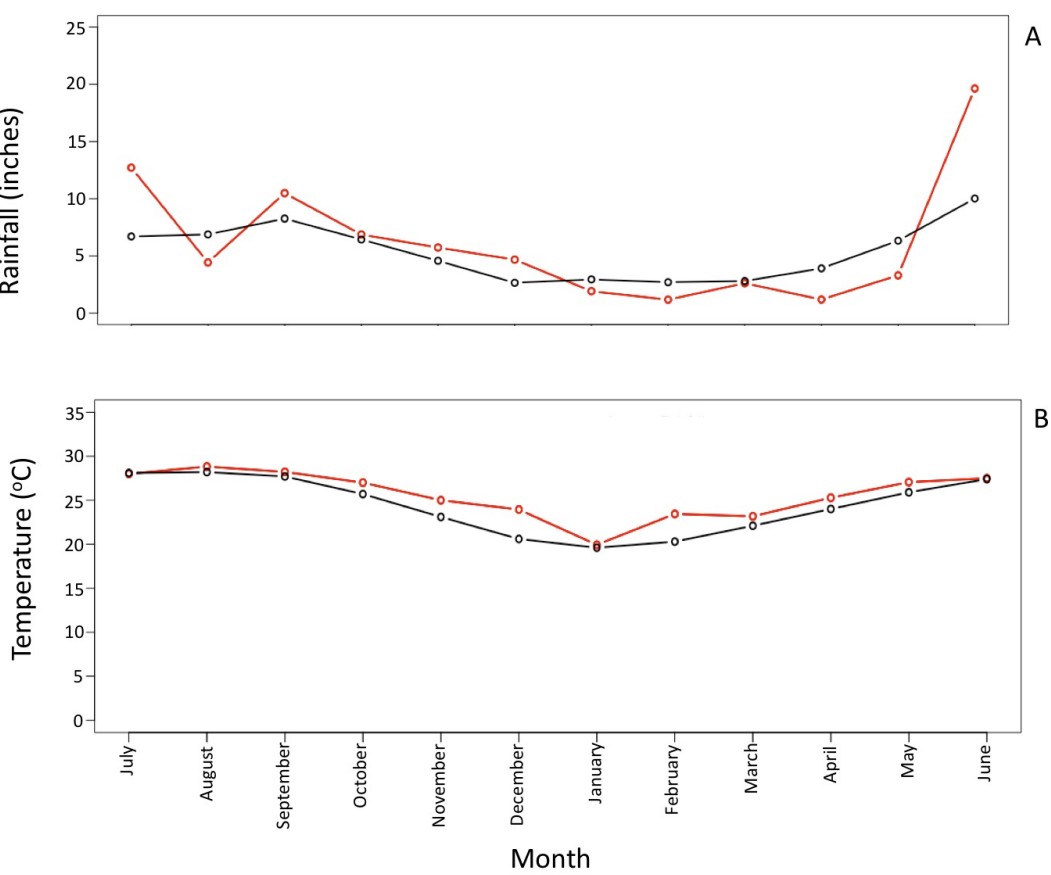

**Figure 4  Average rainfall and temperature values for PEI from July 2013 to June 2014.** (A) The average rainfall values for PEI and surrounding areas. The red line indicates the average rainfall levels for PEI during the study. The black line indicates the historical average of rainfall levels for PEI and surrounding areas. The black line is comprised data collected from 1981–2010. (B) The average temperature values for PEI. The red line indicates the average temperature values during the current study. The black line indicates the historical average of temperature values for PEI. The black line is comprised of data collected from 1981–2010. The historical averages were obtained from: https://www.ncdc.noaa.gov/cdo-web/datatools/normals.

present, but not in similar abundances. This was not unexpected. The wet season in Florida is characterized by increased rainfall that can result in more runoff into the inlet. August had the lowest levels of species evenness and average levels of species richness. August also significantly differed in its microbial community composition when compared to July and September, which did not differ significantly in their bacterioplankton composition. The lower than average rainfall levels in August may have led to decreased sample diversity. Beta diversity of PEI samples was examined and showed that community composition fluctuations were seen between months, seasons, weather at time of sampling, and that there were strong correlations between fluctuations in community composition relating to salinity levels and water temperature. The difference in beta diversity due to weather at time of sampling (rain vs. no rain) was most likely due to surface water dilution during sampling resulting in fewer bacterioplankton organisms sampled during times with rain. Taxonomic

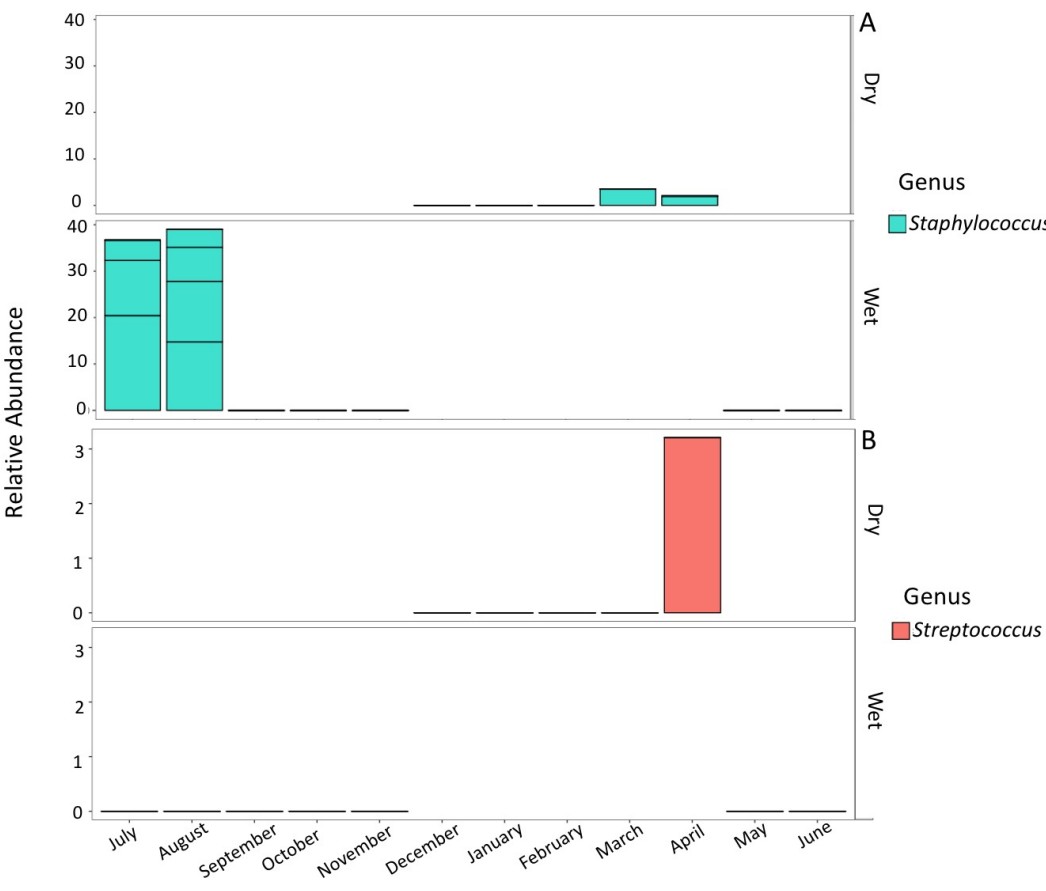

**Figure 5 Comparison of relative abundance levels of potentially pathogenic genera in PEI.** (A) Stacked bar chart displaying the prevalence of *Staphylococcus spp*. between wet and dry season. Each box comprising the bars in the plots is a different OTU. (B) Stacked bar chart displaying the prevalence of *Streptococcus spp*. between wet and dry season. *Staphylococcus spp*. were present in much higher relative abundance levels than *Streptococcus spp*. as displayed by the different *y*-axis scales.

classification of sequences at the phylum level yielded similar bacterioplankton community composition results to previously completed coastal seawater studies (*Campbell et al., 2015*; *Gifford, Sharma & Moran, 2014*; *Elifantz et al., 2013*; *Rappe et al., 2002*). The most evident microbial abundance shifts are seen at lower taxonomic levels. Interestingly, an inverse trend in abundance levels was observed between Proteobacteria and Cyanobacteria throughout the year. It is unknown what caused this trend and more research needs to be completed to determine if these taxa are impacting each other's growth and abundance levels, or if it is other environmental factors causing this trend.

*Synechococcus*. The genus *Synechococcus*, decreased in relative abundance during the winter months, or dry season, and increased in abundance during the summer and early fall, or wet season months. *Synechococcus spp*. are in the Cyanobacteria phylum. Our data correlate with previous observations of increased cyanobacterial blooms in Florida coastal and freshwater ecosystems in the late summer and early fall months. The blooms are caused

by warm water conditions paired with increased sunlight levels, and nutrient loading from urban runoff (*Flombaum et al., 2013*).

**Thalassiosira**. Another taxon seen to fluctuate in abundance with the seasons was the diatom *Thalassiosira spp.* Previous studies have shown that this genus is a common resident in marine coastal surface waters, and tends shift in abundance on a seasonal cycle, increasing in abundance with spring algal blooms (*Heinzelmann et al., 2016*). Our results indicate that this organism increased in abundance during dry season months, however it was seen at its highest relative abundance in early spring. While it would be expected that *Thalassiosira spp.* should increase in abundance during the wet season, it was present in highest abundance in the month of April, the month which marks the start of the wet season.

**Rhodovibrio**. Interestingly, members of the genus *Rhodovibrio* were seen only in samples in Florida's wet season. *Rhodovibrio* is a genus commonly seen in oceanic biofilms or microbial mats. There have not been many studies completed on ecosystem roles of *Rhodovibrio*, although it is known that this organism is classified as a purple non-sulfur bacterium which can secure energy through light by being a photoorganoheterotroph or can secure energy through organic matter by being a chemoorganoheterotroph (*Stal, Bolhuis & Cretoiu, 2017*). The Florida wet season spans summer months when typically, more light availability and warmer water temperatures occur. These conditions could have stimulated biofilm formation and proliferation which could explain why members of the *Rhodovibrio* genus were present in higher abundance during the wet season.

**Acholeplasma**. Fluctuations in abundance were seen in the genus *Acholeplasma*. Microbes in this genus are in the class Mollicutes and are characterized by lacking a cell wall. Microbes in this genus are known to be commensal organisms in plants, insects, and vertebrates (*Kube et al., 2014*). It is unlikely that microbes from this taxon are residents of the surface water microbial community, and most likely the presence of this genus in PEI samples is due to size selection when filtering, or because this is the closest classified organism at 97% sequence similarity in the RDP database.

## Potential pathogens

PEI represents a point source of pollution introducing harmful pollutants into the surrounding marine environments including the Florida coral reef tract and recreational beaches (*Banks et al., 2008*; *Stamates et al., 2013*). Due to the influence of the inlet waters on the surrounding marine environments, it is important to examine the presence of potentially pathogenic organisms in the inlet waters.

The Bacilli class contained two known potentially pathogenic genera *Staphylococcus, and Streptococcus. Staphylococcus* abundance levels were highest in the months of July and August; however, members of this genus were also present in March and April. *Staphylococcus* is a genus of gram-positive bacteria commonly found on the nails, skin, and hair of humans (*Lian et al., 2012*). This taxon can thus be shed directly into coastal waters from bathers. The well-known species in this genus, *S. aureus*, can also cause illness in humans. *S. aureus* has a high resistance to salinity, making it a potential threat to other humans using the contaminated water source for recreational purposes. While this species

commonly links to both human symbiosis and illnesses, marine mammals have also been infected (*Van Elk et al., 2012*; *Bik et al., 2016*). The origin of the strain of *S. aureus* that is contracted by marine mammals was most likely from terrestrial sources introduced into the marine environment via runoff (*Van Elk et al., 2012*). Studies examining the abundance of *Staphylococcus* over a wet and dry season at a heavily visited coastal area observed increased abundance of *S. aureus* during the wet season (*Curiel-Ayala et al., 2012*). These data also showed the highest increased abundance of *Staphylococcus* during Florida's wet season.

Streptococcus sp. were present in highest abundance in April. Increased freshwater input and warm water conditions could have been the cause of the increased abundance of these organisms. This presence also coincided during the prime shipping season in Port Everglades, which may have had an impact on abundance levels of *Streptococcus spp.*

Enterococcus spp. are important fecal indicator bacteria, most often utilized to assess fecal contamination on recreational beaches and coastal areas (*Aranda et al., 2015*; *Heaney et al., 2014*; *Wade et al., 2003*; *US Environmental Protection Agency, 1986*; *US Environmental Protection Agency, 2004*). A recent study examining the number of exceedances of enterococci on recreational beaches in Miami-Dade County, FL from 2000–2010 (*Aranda et al., 2015*) showed that beaches were only in exceedance of the allowable levels of enterococci 3% of the time. This study examined data generated by the Florida Healthy Beaches Program, which samples weekly. No patterns in regard to rainfall or storms were seen in correlation with enterococci exceedances, although this may be due to the sampling frequency and high decay rate of enterococci in marine waters (*Aranda et al., 2015*). In contrast to this, a study completed by Curiel-Ayala and colleagues in Mexico (2012), showed increased *Enterococcus* levels during the rainy season, and the highest concentrations of the genus corresponding to highest tourist presence. The highest levels of enterococci seen in the Miami-Dade study were in March and October, which could be due to high tidal levels in October, and possible tourism influences in the month of March overlapping with spring break (*Aranda et al., 2015*). It was expected that presence of *Enterococcus spp.* would be in highest abundance in PEI during March-April when shipping season and tourist season start. Interestingly, *Enterococcus spp.* were not observed in any of the samples taken in PEI. This was also seen in the previous study examining presence of pathogens in PEI (*Campbell et al., 2015*). It is possible that *Enterococcus spp.* are not prevalent inhabitants in PEI. More intensive studies would need to be completed to determine if there is the presence of *Enterococcus spp.* in PEI.

It was interesting that the causative agent of the white pox disease for coral *Acropora palmata,* human fecal bacteria *S. marcescens* was not observed in the top 100 OTUs in the Enterobacteriales class. A recent high throughput molecular study of coral white band disease identified only five orders with large numbers of disease-associated OTUs: *Flavobacteriales, Alteromonadales, Oceanospirillales, Campylobacterale* and *Rhodobacterales* (*Gignoux-Wolfsohn & Vollmer, 2015*). Also of note is the absence of *Vibrionales* pathogens in our dataset. The previous analysis of PEI microbiomes by *Campbell et al. (2015)* showed a low abundance of *Vibrionales.*

While it is intriguing that potentially pathogenic genera were detected in PEI, it is not possible to conclude that these organisms are virulent without culturing, in depth

metagenomic sequencing, or oligotyping analyses (*Eren et al., 2013*). A general pitfall of 16S amplicon sequencing is the lack of taxonomic resolution down to the species and strain level. Without this level of classification pathogenicity cannot be confirmed. A previous 16S amplicon and shotgun sequencing study completed on the microbiomes present in the subway system in Boston, showed that there were very few antibiotic resistant genes and virulence factors found in the samples. They also found that there were very few sequences that were identified as opportunistic pathogens or potentially pathogenic organisms (*Hsu et al., 2016*). This suggests that most likely the microbes identified in this study did not contain virulence factors and are non-pathogenic commensals living in the inlet. However, future research utilizing deeper sequencing and culture-based methods should be completed to further examine presence of pathogens in PEI.

## SIGNIFICANCE AND CONCLUSIONS

This study has provided one of the first in-depth profiles of the bacterioplankton community in metropolitan S. Florida inlet waters. The current dataset complements and expands upon a previous a pilot HTP study characterizing the bacterioplankton community of Port Everglades inlet and surrounding coastal waters. Specific marine habitats, such as coral reefs or mangroves, have well defined optimal conditions for thriving and can be sensitive to small perturbations (*Precht & Miller, 2007*; *Hoegh-Guldberg et al., 2007*). The data from this study will be helpful to local environmental managers, such as the Southeast Florida Coral Reef Initiative (SEFCRI, http://www.dep.state.fl.us/coastal/programs/coral/sefcri. htm), which aims to protect and monitor S. Florida reef habitats. The timeframe of this study appeared poignant, showing that anomalous weather patterns (dry August) can affect community composition. This study also preceded a protracted 2014–2016 El Niño event outbreak of coral disease on the Florida reef tract in 2014 and the planned expansion and deepening of PEI in the upcoming years. Understanding the microbial composition in the inlet prior to these perturbations will contribute to better management and mitigation of the environment and may help to protect the adjacent coral reef where the inlet outflows.

### Funding

The work was supported by a President's Grant through Nova Southeastern University. The funders had no role in study design, data collection and analysis, decision to publish, or preparation of the manuscript.

### Grant Disclosures

The following grant information was disclosed by the authors:
Nova Southeastern University.

### Competing Interests

The authors declare there are no competing interests.

## Author Contributions

- Lauren O'Connell conceived and designed the experiments, performed the experiments, analyzed the data, contributed reagents/materials/analysis tools, prepared figures and/or tables, authored or reviewed drafts of the paper, approved the final draft.
- Song Gao analyzed the data, contributed reagents/materials/analysis tools, prepared figures and/or tables, authored or reviewed drafts of the paper, approved the final draft.
- Donald McCorquodale contributed reagents/materials/analysis tools, approved the final draft.
- Jay Fleisher analyzed the data, contributed reagents/materials/analysis tools, prepared figures and/or tables, approved the final draft.
- Jose V. Lopez conceived and designed the experiments, contributed reagents/materials/-analysis tools, prepared figures and/or tables, authored or reviewed drafts of the paper, approved the final draft.

## DNA Deposition

The following information was supplied regarding the deposition of DNA sequences:

The Port Everglades Inlet Microbiome sequences have been deposited in the NCBI database under BioProject accession number: PRJNA413618 and under the Sequence Read Archive (SRA) accession IDs of SRR6147998–SRR6148078.

## Supplemental Information

Supplemental information for this article can be found online at http://dx.doi.org/10.7717/peerj.4671#supplemental-information.

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
