# Peer review of "Fine grained compositional analysis of Port Everglades Inlet microbiome using high throughput DNA sequencing"

_PeerJ, doi:10.7717/peerj.4671_

## Round 0.1 · original submission · Major Revisions

As pointed out by all reviewers, the manuscript contains a number of flaws that need to be addressed. Please pay particular attention to the comments of Reviewer 1, who provided the most extensive and critical review. Do not assume that you only need to "fix the figures," as suggested by Reviewer 3.

Reviewer 1 ·

Basic reporting

Insufficient background and context is provided for this study. Hardly any data are actually presented.

Experimental design

It is unclear how this study addresses a new (or even valid) research question as presented. Insufficient methodological detail is provided including basic information regarding which hypervariable region was sequenced.

Validity of the findings

Very little data is presented and the interpretations and conclusions of what is presented are not well validated.

Additional comments

The authors characterize the microbial community in an inlet by taking weekly samples for a year. The study is poorly presented, the data are not clearly shown, and the context and conclusions stated are inappropriate given the existing body of knowledge regarding aquatic microbial communities. It is unclear what hypervariable region was sequenced, the data are not publicly available, and it is not clear how the authors leveraged a weekly sampling strategy since hardly any data are actually shown.

Broadly, the abstract is a bit long, although within the journal guidelines, but could be made more clear by condensing general descriptions and adding in more statistical values. Discussion of phyla is not very informative. In contrast, the introduction is relatively brief and should be expanded to provide more literature, especially regarding next-generation sequencing studies of bacterial communities, to better contextualize this study. The results section is among the strangest I have ever reviewed as nearly no actual data are presented and analyses. This must be corrected. Furthermore, it is unclear why the authors are broadly evaluating communities at Phylum and Class levels, which are poorly informative and lend no value beyond what already exists in the literature. Similarly, the discussion has no focus and requires more specific direction and organization.

Specific comments
Abstract:
line 34: Please remove the sentence, "Over 1.4 million 16S rRNA V4 reads generated a total of 16,384 Operational Taxonomic Units (OTUs) from the PEI habitat". This is too minute a detail for the abstract.
line 37: Italicize Bacillus.

Introduction
lines 87-91: This is a rather long and confusing sentence. Suggest splitting into two.
line 94: Please revise as "next-generation sequencing characterization" or something more accurate. As currently written, I believe this overstates the scope of the characterization.
line 100: Please expand the introduction to provide some rationale for these hypotheses.
line 106: Please revise this final sentence. While this study does differ from those cited, there are many others studies that have performed NGS characterization of bacterial communities in various aquatic ecosystems.

Methods
The sampling strategy needs to be clarified. It is not clear how weekly sampling for a year at two sites results in 82 samples. This should intuitively be 52 or 104 samples. Additional information is also necessary describing the PCR amplification e.g. primers used, numbers of cycles, any barcoding, etc.
line 116: Correct "taken".
line 120: Correct to "by vacuum filtration".
line 136: Remove "raw". The first two sentences here are also redundant.
line 140: At what similarity were OTUs called?
line 159: Why is regression analysis treated separately from other statistical analyses? And what is the justification for a = 0.10 here?
line 164: These should be referred to as POTENTIAL pathogens.

Results
line 170: 82 samples were stated in the results, but there are 151 here. Please clarify.
line 174: Were samples rarefied to account for different numbers of reads between samples?
line 182: Regarding taxa present in >1% of samples, of course these are the most abundant. What information is this sentence meant to convey?
line 183: Why are none of the alpha diversity indices presented? Where is this data? And how are these indices not significantly different? Chao1 values are orders of magnitude greater than Simpson or Shannon indices.
line 187: Please revise this sentence to be more clear. Was August diversity greater or less than other months?
Figure 1: Please revise so that some taxonomic classification is presented along the x-axis. Otherwise, this figure is relatively meaningless.
line 196: The use of this 1% cutoff is very strange. Why not talk about actual abundances of taxa?
line 197: This sentence belongs in the discussion.
line 205-209: This is methods.
line 220: Please correct "than"
line 220-221: methods
line 227: Corrections for multiple comparisons should be performed when interpreting these results.
line 232: Why were analyses performed at class level? More specific taxonomic characterizations should be used to allow better interpretations of results. The entirety of this paragraph is also otherwise shown in the table. Please revise.
Chemical data: Why are these data not presented?
line 234-235: This description is not necessary.
lines 251-266: If the authors mean to address differences in relative abundances, these differences should be assessed statistically.

Discussion
line 277: What was the basis for this expectation?
line 282-284: It is unclear how this idea is connected to the rest of the paragraph.
line 287: Citing of unpublished sources does not provide a valid reference.
line 291: This statement is not accurate. The current study cannot confirm comparisons with other sample types that were not evaluated.
line 292: But the beginning of the discussion supposes that samples were not marine. Please clarify or explain.
line 309: If the authors wish to discuss rainfall data, it should be presented in the results.
line 339: What is the "microbial loop"? Please clarify.
Line 409-410: This statement is not accurate. The authors cite previous studies from their own lab as well as Staley, 2017 which previously characterized these commmunities. Please remove this sentence.

Reviewer 2 ·

Basic reporting

Overall, the article is well-written and structured, most citations are appropriate, the raw data is available, and the article is self-contained. However, the following concerns should be addressed:

1. The introduction, especially the second paragraph, seems to be focused more on biogeochemical cycles and only describes potential pathogens in the marine system in one line. However, at least a third of the discussion is only on potential pathogens found in the data set. The introduction would be improved by adding more context in this regard.

2. While the majority of citations are appropriate, I find it odd that on line 78, three citations are used to show that a reef is directly offshore from the sampling site. I would think only one would be necessary or perhaps a map without any citations? In addition, on line 107, the study by Carsey et al is attributed to the year 2012, while in the reference list, this study is said to be in preparation.

3. The article is well-written and easy to follow overall, but there are a few inconsistencies in style and a few grammatical errors, including:
a. line 113, L does not need to be defined as it is the S.I. abbreviation
b. line 115, in situ should not be hyphenated and should be italicized
c. line 122, 'gel was run' is technical jargon
d. line 152, Kruskal-Wallis should be capitalized as it is in other mentions in text
e. lines 202 and 335, Portiera should be capitalized
f. line 237, there is a space needed between Flavobacteriia and (R2=0.21)
g. from line 210 – 250, p-values are sometimes reported as P, others as p, and others still as italicized p. I realize that different style guides suggest any of the above for p, but they should be consistent throughout the text.
h. line 277, Is ICW defined elsewhere?

Experimental design

The experimental design is sound, and the methods are detailed sufficiently for others to replicate the study except for a few areas, which are detailed below.

1. The primers used to amplify 16S rRNA genes should be explicitly stated. While the Illumina Sequencing Library Preparation guide does detail which primers are used, it leaves the readers to first consult this guide and then to consult the Klindworth et al paper for a full description of the primers used in this study. I would suggest directly citing the Klindworth paper.

2. The current study uses cd-hit to assign OTUs but fails to state which cut-off is used, which is up to the user when using cd-hit. Was the default (97%) cut-off used?

3. What was the justification used for having an alpha level at 0.10 for the Multiple Least Squares Regressions? Few of the significant results would be altered by using the more standard alpha level of 0.05. Even with the current alpha level, the statement that the relationship between Flavobacteriia and salinity is false as the stated p-value of 0.101 is above the alpha level used.

Validity of the findings

The findings and conclusions are logically stated and are borne out by the data and statistical analysis. I only have one issue that require clarification.

1. The present study refers to Chloroplast as a class. As far as I am aware, Chloroplast is an organelle and not a class of bacteria, which the text seems to make it out to be. Does this class designation refer to all 16S sequences that were assigned to belong to various chloroplast genomes? Please clarify what is meant by class Chloroplast and do these sequences annotate as specific types of eukaryotic algae.

·

Basic reporting

The language was clear and unambigous.
References were clear and correct.
The figures are not great and not very clear what they are trying to show. They need much more work in order to illustrate the message clearer.
I couldn't find a link or other to the SRA or other repo where the data is.
I couldn't really tell from the data presented that there was a difference between wet and dry - at least as presented. Especially from the NMDS plot.

Experimental design

These were fine. The plots need some cleaning up. But, the methods were will defined.

Validity of the findings

Once again, the plots really are confusing and don't say much relating to your message. I think it is there it just needs some more clear figures to show it.

Additional comments

Fix the figures to match the text better and you should be good to go.

---

## Round 0.2 · Major Revisions

The manuscript is not yet up to standard. The 3 reviewers have provided additional useful comments. The number and impact of these comments required me to still classify the revisions needed as 'major'.

Reviewer 1 ·

Basic reporting

The authors have significantly improved this manuscript upon revision.

I suggest the number of figures be reduced by combining Figures 1 and 2 into two separate panels of the same figure. Figures 4 and 5 as well as 6 and 7 could be similarly combined.

Finally, the raw data need to be made available in the Sequence Read Archive at NCBI or a similar depository and an accession number should be provided.

Experimental design

The description of the potentially pathogenic bacteria is still extremely difficult to interpret. As noted in specific comments, how these analyses were performed needs to be clarified since it is unclear in the revision which taxonomic level was being interrogated.

Validity of the findings

The authors would like to claim a fine-grain sampling strategy on the basis of weekly sampling, but the data presented represent only a monthly survey of the microbiota. As such, the authors need to clarify this, including revision of the title, to reflect the nature data presented.

Additional comments

Specific comments:
line 30: Change "will be" to "were".
lines 32-33: Monthly sampling is not uncommon, so the statement that this represents a higher frequency or finer scale than other studies is misleading.
line 40: What is meant by "significantly high abundances"? How was this level of significance determined? please also clarify at line 52.
line 49-51: The statement about N and P should be omitted unless this was evaluated statistically.
line 53: Suggest deleting the sentence starting "To our knowledge...".

lines 109-110: It seems a word is missing here.
line 111: The abstract states that monthly samples were analyzed, but weekly is stated her. Please clarify.

line 135-136: Please provide additional information how quality was determined. e.g., Did the samples that were dropped have low concentrations of DNA?
line 155: Should this be updated to QIIME 2?
line 178: Please provide a reference for the ANCOM test. (Mandal S, Van Treuren W, White RA, Eggesbø M, Knight R, Peddada SD (2015) Analysis of composition of microbiomes: a novel method for studying microbial composition. Microb Ecol Health Dis 26:27663)
line 182: Spell out PCoA at first use.
lines 193-200: This section is confusing as it appears genera were picked based on evaluation of order abundances. There is also the mention of OTUs. Which taxonomic level was evaluated?

line 255: Please specify "genera" instead of "taxa".
line 266: What is meant by "lower than expected"?
line 270-1: This sentence is redundant with the methods. Please delete.
line 278: Please specify "potential" pathogens.

line 325-369: Please break this section up into several paragraphs. I also suggest deleting the discussion of Rhodovibrio and Acholeplasma since the information presented is merely a description of the genus and is unrelated to the current study.

Reviewer 2 ·

Basic reporting

The manuscript is much improved from the initial submission. I appreciate all the work that was done to answer the questions drawn by the reviewers. While it has improved, I do have a few issues that I feel should be addressed before publication:

1. In the abstract, it says that the samples were collected monthly, however, in text, the samples were stated to be collected weekly.

2. The text states that 82 samples were sequenced, but the submission to the NCBI short read archive only has 81 different runs. Was one run poor quality or just lost in the fray during submission? I understand that not all of the runs were used for analysis.

3. Table 1 is formatted in such a way that makes it difficult to read. The OTU names are particularly useless. I'm not sure what it is supposed to be selling me on in its current iteration.

4. In figures 6 and 7, I assume that the historical means used are from the average of every month between 1948 and 2000 as they are flat lines. Is this really the best thing to compare monthly data to? Was this a mistake and was the average from each month supposed to be presented instead? Or is the average temperature and rainfall from month-to-month the same?

Experimental design

No comment.

Validity of the findings

No comment.

·

Basic reporting

The text looks much better. But, the pathogenic bacteria is a tough one for me. I would dial back the language on that a bit.
I think a plot on the pathogenic genera at the strain level would be very helpful. And, some brief language on how MiSeq isn't perfect and a reference to curtis huttenhower paper debunking the pathogenic bacteria in the subway.
I think it would be good to except with that infomation and some more cleaning of the figures/tables.
If they are fixed and I can't see them let me know.

Experimental design

The methods sections are greatly improved.
Yes, I think this section is up to par.

Validity of the findings

This was fine.

Additional comments

If they are fixed and I can't see them let me know.
Table 1 - use OTU numbers like OTU1 etc with the taxonomic id. These numbers as presented are useless and makes it hard to review or better yet plot this!

Table 2
The words are running down on the next line. plot or fix

Figure 1 and 2
Please replot in ggplot2 and facet it to combine into one figure. Also make the y-axis larger

Figure 3
remove grey background - theme_bw(), would look much cleaner

Figure 4 and 5
Instead of stacked use a side by side taxa on the y-axis for the phyla it would be easier to read. The colors need to be fixed as well.

Figure 6 and 7
Please replot in ggplot2 and facet it to combine into one figure. Remove the default excel plot for a fresher and cleaner look.

Figure 8
Instead of stacked use a side by side taxa on the y-axis or facet it. The pink is a terrible color.

Figure 9
Same as figure 8 remove purple. However combining figure 8 and 9 would be better and a barplot on the overall abundance of bacillus including a narrow look at strains of straph and bacillus pathogens.

I think if these could be fixed then it's okay for me.

---

## Round 0.3 · accepted · Accept

I would like to thank the authors for choosing PeerJ.

# Reviewer 1 ·

Basic reporting

All comments have been successfully addressed.

Experimental design

All comments have been successfully addressed.

Validity of the findings

All comments have been successfully addressed.

Reviewer 2 ·

Basic reporting

No comment

Experimental design

No comment

Validity of the findings

No comment

Additional comments

The current iteration of the manuscript represents a marked improvement from the initial submission. The wording is much more clear, the rationale behind the study is evident, the methods are sound, and the findings are presented and interpreted in an appropriate manner. I have no further comments on what could be improved.

I would like to thank the authors for taking into careful consideration the comments from myself and the other reviewers.